# Combining Graph and Recurrent Networks for Efficient and Effective Segment Tagging [*]

**David Montero**
NielsenIQ, Spain
david.montero@nielseniq.com

**J. Javier Yebes**
NielsenIQ, Spain
javier.yebes@nielseniq.com

## Abstract

Graph Neural Networks have been demonstrated to be highly effective and efficient in learning relationships between nodes locally and globally. Also, they are suitable for documents-related tasks due to their flexibility and capacity of adapting to complex layouts. However, information extraction on documents still remains a challenge, especially when dealing with unstructured documents. The semantic tagging of the text segments (a.k.a. entity tagging) is one of the essential tasks. In this paper we present SeqGraph, a new model that combines Transformers for text feature extraction, and Graph Neural Networks and recurrent layers for segments interaction, for an efficient and effective segment tagging. We address some of the limitations of current architectures and Transformer-based solutions. We optimize the model architecture by combining Graph Attention layers (GAT) and Gated Recurrent Units (GRUs), and we provide an ablation study on the design choices to demonstrate the effectiveness of SeqGraph. The proposed model is extremely light (4 million parameters), reducing the number of parameters between 100- and 200-times compared to its competitors, while achieving state-of-the-art results (97.23% F1 score on CORD dataset).

## 1 Introduction

Information Extraction (IE) has become a focus task over the last years within the Machine Learning community. There is a growing need to automate the extraction and storage of information from documents, especially from unstructured ones. The rise of Deep Learning has been extremely beneficial, leading to the release of models capable of performing with the same quality as humans [1–5]. Moreover, a myriad of businesses and use cases within industry require the automatic processing and understanding of documents and their contents to convert unstructured data into their semantically structured components. The main aim is to reduce the manual burden in day to day operations implementing efficient and cost-effective automatic solutions.

Within IE, the semantic tagging of the text segments (a.k.a. entity tagging) is an essential task that allows the system to understand the different parts of the document and to focus on the most relevant information. Usually, these text segments are detected in a previous stage by an OCR engine at word level. This paper presents an innovative solution for efficient and effective segment tagging on unstructured documents.

Existing entity tagging models are huge, they contain an overwhelming number of parameters (hundreds of millions) that could be highly reduced. Furthermore, most of them are purely or mostly based on Transformer [6] architectures [1, 3–5]. They need to define a sequence limit and, therefore, they suffer from the sequence truncation problem. In addition, this sequence limit must be chosen carefully, as the computational complexity increases quadratically with it [7]. This is due to the fact that the Transformers are fully connected architectures, where each segment needs to interact with the rest.

---

[*] A patent has been applied for that covers the subject matter described in this article.

D. Montero et al., Combining Graph and Recurrent Networks for Efficient and Effective Segment Tagging [†]
*Proceedings of the First Learning on Graphs Conference (LoG 2022)*, PMLR 198, Virtual Event, December 9–12, 2022.

These challenges can be solved using more flexible architectures, such as the ones based on Graph Neural Networks (GNN) [2, 8, 9]. GNNs have been demonstrated to be highly effective and efficient in learning relationships between nodes locally and globally [9, 10], as well as between nodes of different types [11, 12]. However, they do not use the sequential order of the nodes as a source of information, which is important for the considered task. To this extent, some attempts have been made to compensate for this limitation by combining them with other mechanisms, such as recurrent layers [2]. We believe they are also overparameterized and that the selection and treatment of the features is not optimal. To overcome these limitations, we present SeqGraph, a new model that combines Transformers for text feature extraction, and GNNs and recurrent layers for segments interaction, for an efficient and effective segment tagging. The main contributions of this work are:

- Extremely light-weight entity tagging model (4 million parameters) capable of achieving state-of-the-art results (97.23% on entity tagging task of CORD dataset [13]).

- Optimal selection and extraction of the node features from the text and bounding boxes of the segments.

- Optimized model architecture combining Graph Attention layers (GAT) [14] and Gated Recurrent Units (GRUs) [15].

- Ablation study on the impact of the different sources of information on the model accuracy and parameters.

## 2 Related Work

In recent years, the increase in the demand for information extraction systems has been reflected in the number of publications and, consequently, also on entity tagging [1–5, 8, 9, 16–22]. Although there are few works that attempt to solve the problem from scratch [23], almost all the released models rely on text segments extracted using an OCR engine. Most of them are based purely or partially on Transformer architectures [6]. However, there is an emerging trend on the application of GNNs for entity tagging.

### 2.1 Transformer-based models

Since the most basic versions, such as BERT [24] or RoBERTa [25], which only use the text and the sequential order of the segments to extract the input features, a lot of novelties have been introduced to enhance the Transformers performance on this task. New models include different sources of information, such as the image or the layout, but also new ways of extracting the features and combining them. For instance, in several works, the authors inject the layout information of each segment into their features, some of them at word level [1, 16–19], and others dividing the document into regions that share the same embeddings [3, 5, 20, 21]. In some cases, the layout information has also been used to enhance the self-attention mechanisms, usually as a bias term [3, 4, 16, 22, 26]. The image features are usually integrated with the textual ones by concatenating or adding them [5, 18, 19, 22], but some models use more sophisticated ways of combining them. For instance, in [3], [4], and [20] the authors use a multi-modal transformer architecture that incorporates a multimodal self-attention to enforce the cross-modality feature correlation.

### 2.2 GNN-based models

Within IE tasks, GNNs try to overcome the limitations of the Transformer-based models. The Transformers are fully connected architectures, where each segment must interact with each other. The computational complexity increases rapidly with the number of segments [7]. In addition, they need a predefined maximum sequence length, leading to truncation problems for large sequences. GNNs avoid all these problems with their flexible structure, where each segment only needs to interact with a reduced number of neighbors. However, the setup is more complex, with some critical design choices, such as the graph structure, the edge sampling strategy, or the message propagation approach. Some promising approaches have been recently released. For instance, in [8] the authors propose a GNN-based model for solving entity tagging (ET), building (EB), and linking (EL). First, they generate a k-Nearest Neighbor (kNN) graph at text segment level for solving the EB task as an edge prediction task, using features extracted from the bounding boxes and from the text and passing them through several GAT layers. Then, the entity features are computed by aggregating the

output features from the GNN and processing them with a linear layer. The features are used to solve the ET task, using a Multi-Layer Perceptron (MLP) classifier, and the EL task, evaluating all the possible entity pairs with another MLP classifier. In [9], the model incorporates also visual features from the image. The text features are extracted using a BERT encoder and the visual ones using a SWIN Transformer with a Feature Pyramid Network (FPN). Both vectors are enriched with layout information by adding a layout embedding. Then, before each GAT layer of the GNN architecture, the visual features are fused into the input features using a fusion layer. In these layers, the relative layout is also included in the self-attention mechanism. On the other hand, in [2] the image and text features are fused before feeding them into the GNN. The layout information is not embedded into the node features but used for computing the weights for the message aggregation within the Graph Convolution Network (GCN) layers [27]. The text features are extracted using a Transformer encoder at character level and the image features using a Convolutional Neural Network (CNN) architecture. Then, the character features of each segment are averaged and passed through the GNN layers. The output features are then aggregated to the previous features at character level and fed to two bidirectional LSTM layers [26] in order to extract the sequential information. Finally, they use the Viterbi algorithm to generate the final predictions.

All the above works have some drawbacks. In [8], the quality of the extracted text features is poor, and they do not leverage the sequential information, which is important for the considered task. Consequently, the results obtained are very limited. In the case of [2] and [9], the models have a huge number of parameters, in part due to the heavy image backbones that they use. In this work we aim at combining the benefits of all of them to make a light, fast, and effective entity tagging model.

## 3 Methodology

### 3.1 Problem definition

Given a list of text segments (usually at word level) provided by an OCR engine who extracted them from an image of a document, the goal is to tag each segment with its corresponding semantic category from a closed list. Each segment consists of the text string and the rotated bounding box. For instance, having a purchase receipt, each segment could be tagged as store address, phone, date, time, item description, item value, etc. Figure 1 shows an illustration.

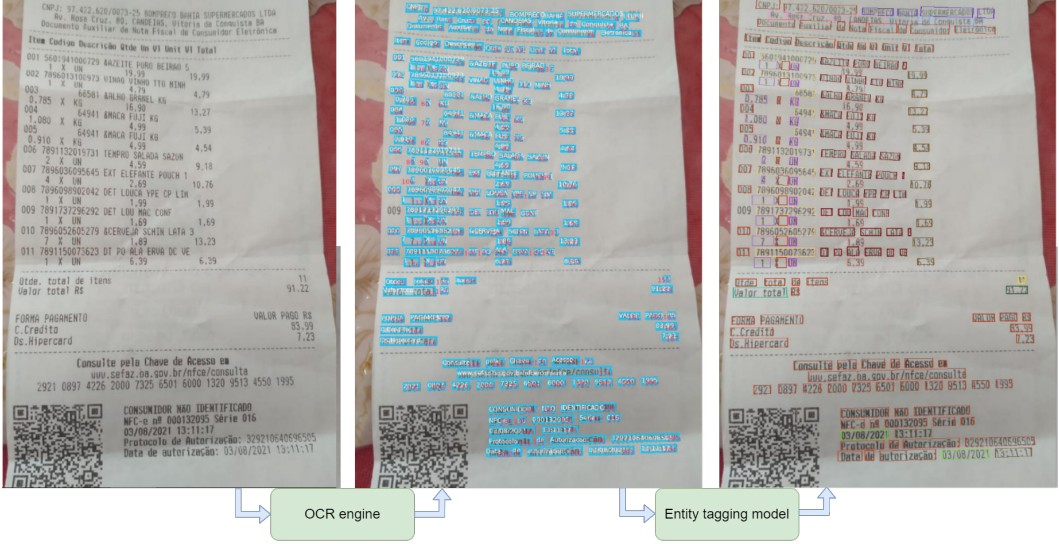

**Figure 1:** Illustration of document entity tagging. In the right image, the color of the bounding box denotes its category. Segments with the bounding boxes of the same color belong to the same category.

Some of the challenges present on the raw collected images of documents include highly unstructured documents, such as purchase receipts, with multiple and complex layouts and non-natural language (abbreviations, brands, product names, punctuations, etc). The noise caused by the OCR engine

(errors in the text, inaccurate bounding boxes, missing or duplicated detections...) and due to bad image and/or physical conditions (perspective, rotation, wrinkles, ripples...).

## 3.2 Overview

Given the above background, we focus on a GNN-based model due to these reasons:

- Graph-based representations are flexible and capable of adapting to complex layouts.

- The task can be defined as node classification, where the semantic category of a node is highly dependent on the features of its neighbor nodes and on the global context. The literature about GNNs has demonstrated they are highly effective and efficient in learning relationships between nodes locally and globally [9, 10].

- The number of nodes in the document varies from a few to hundreds of them, which can be unfeasible to process for fully connected networks or Transformer-based models. However, for this use case, the number of interactions can be limited based on the bounding box coordinates, accelerating the inference and reducing the number of required resources. GNNs are suitable for this type of highly sparse data structure.

We must note one of the weaknesses of the proposed GNN-based approach. GNNs do not consider the position of the segments in the input sequence. However, it is important to keep an order to read a document beyond text layout prediction. Several approaches have tried to overcome this limitation injecting the sequential information into the node features [28], using it within the attention mechanism of the GAT layers [29], or combining the GNN with recurrent layers [2]. The first two require defining how to extract and combine it with the rest of the features, which can be tedious and lead to a more unstable model. In addition, it requires increasing the number of GNN layers or its number of parameters to learn from this new source of information. On the contrary, in recurrent layers this information is learnt directly from the order of the segments with a reduced number of parameters and without altering the GNN architecture.

Following this reasoning, we developed SeqGraph, a hybrid model based on GNN and RNN for text segment tagging as in Figure 2. Starting from the list of text segments coming from the OCR, SeqGraph first extracts and preprocesses the text and region features from each segment. In parallel, it generates the segment nodes and performs the edge sampling between them. Next, the node features are passed through the graph attention layers and get enriched by their neighbors. Two bidirectional Gated Recurrent Unit (GRU) layers [15] processes the featuers to add information about the order of the segments. Finally, we add a linear layer and a Softmax layer to obtain the output probabilities for each text segment.

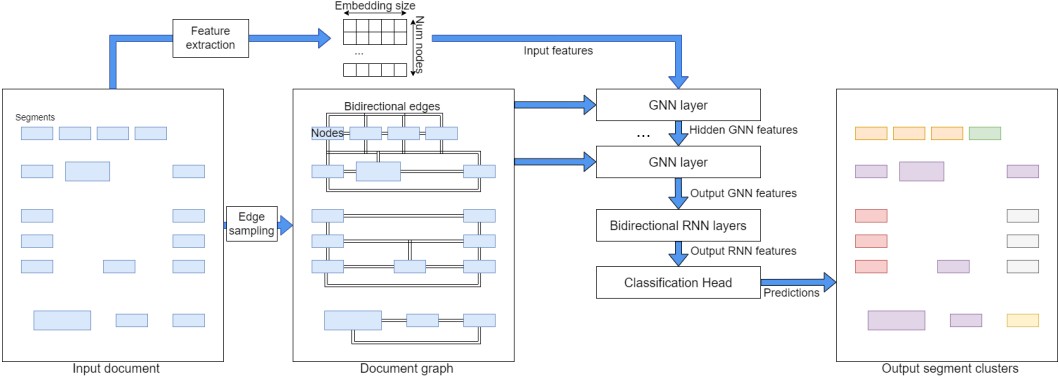

**Figure 2:** High Level Architecture of the proposed model.

## 3.3 Feature extraction

We use three sources of information: the text string, the bounding box and the position in the sequence.

Diving through the literature, we can find different approaches for extracting features from the text of a segment, but we can group them into two categories: the ones that extract the features attending to its semantic meaning and the ones that extract the features attending to its composition. The first one assigns a feature vector to each text string (usually at word level) using an embedding layer and a predefined dictionary [8, 30]. The embedding layer can be pretrained on another dataset or it can be trained directly from scratch, using the training set for generating the dictionary [8]. The latter method, extracting the features attending to the text composition, means inspecting its characters and their position within the text and finding relevant relationships between them [2]. We include a deeper dive with pros and cons of each of the two approaches in the Appendix A.1.

Analyzing both approaches and the challenges of the segment tagging task for documents, we adopt the second approach, text feature extraction based on its composition, similar to how it is done in [2]. We decide to consider only ASCII characters, setting the length of the dictionary to 128. We convert all the Unicode characters using the standard Unidecode Python package. Its function $unidecode()$ takes Unicode data and tries to represent it in ASCII characters using transliteration tables. The characters that cannot be converted are removed. The size of the embedding layer is 256. The Transformer has 3 layers with 4 heads and an internal dimension of 512.

Regarding the rotated bounding box, we select the following features:

- Left center coordinates: middle point between the top-left and the bottom-left vertices of the rotated bounding box.
- Right center coordinates: middle point between the top-right and the bottom-right vertices of the rotated bounding box.
- Bounding box rotation: angle of the bounding box in radians, between -PI/2 and PI/2.

Note that we discard the height of the bounding box as we observed that the model tended to overfit using this feature. We believe that the height of the segment is not a crucial feature for this task, as it might vary across segments that share the same category, and it does not contain reliable information about the distance between different text lines.

Finally, the position in the sequence is already implicit in the order of the segments and used by the recurrent layers. It could be also injected into the node features by using for instance a positional embedding, but that would require selecting a maximum position and truncating the sequences that exceed this length, which would yield a drop of accuracy. In addition, the positional embeddings do not work well with very long sequences, and we want to consider lengths of hundreds. For these reasons, this information is not injected into the node features.

After extracting the textual and bounding box features they are fused by increasing the dimension of the bounding box features using a linear layer to match the textual features one (256) and adding both. The whole feature extraction process is described in Figure 3.

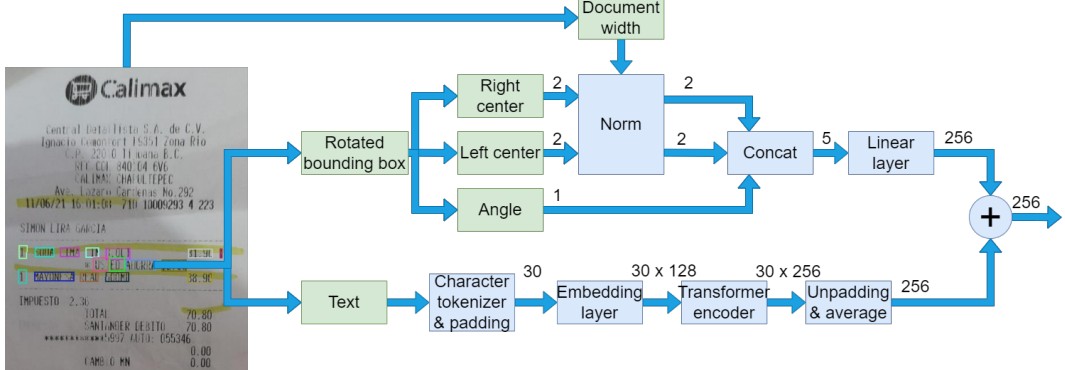

**Figure 3:** Feature extraction process applied to each text segment.

### 3.4 GNN

The GNN architecture takes advantage of the fact that all the information needed for computing the message passing weights (positional and textual information) is already embedded in the node

features and we select Graph Attention Layers (GAT) [14] as the one that best suits our needs. In the GAT layers, the weights for the message passing are computed directly inside the layer using the input node features, in a similar way as it is done in the original attention layers (see Equation 1, where $h$ denotes the node features, $W$ is the weight matrix, and $a$ is the weight vector for the dot product).

$$h_i^{(l+1)} = \sigma \left( \sum_{j \in \mathcal{N}(i)} \alpha_{ij}^{(l)} W^{(l)} h_j^{(l)} \right)$$

$$\alpha_{ij}^{(l)} = \frac{exp(e_{ij}^{(l)})}{\sum_{k \in \mathcal{N}(i)} exp(e_{ik}^{(l)})} \tag{1}$$

$$e_{ij}^{(l)} = LeakyReLU(a^{(l)^T}(W^{(l)}h_i^{(l)} || W^{(l)}h_j^{(l)}))$$

They have been widely applied in document understanding tasks [8, 9]. To avoid 0-in-degree errors (disconnected nodes) while using the GAT layers, we add a self-loop for each node, i. e. adding an edge that connects the node with itself.

The proposed architecture in Figure 4 is composed of 3 GAT layers. All the layers are followed by SiLU activations [31] except for the last one. In our research, this activation worked better than ReLU and other variants. We also add residual connections in all the layers to accelerate the convergence.

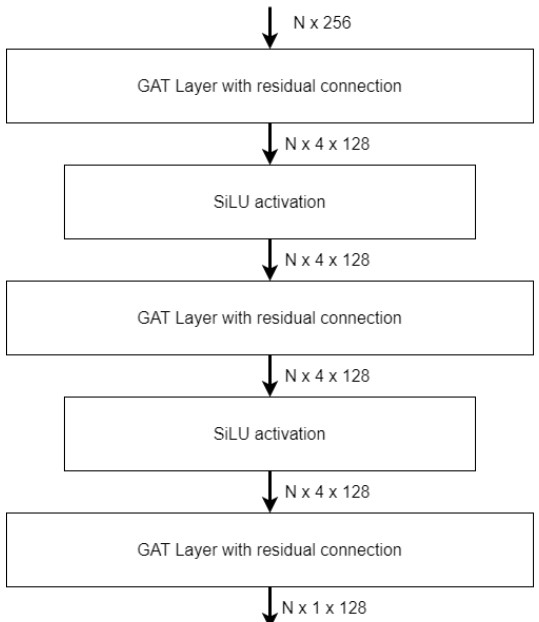

**Figure 4:** Proposed GNN architecture.

Inspired by [9], we introduce a global document node. We use one global node per graph level, and we connect it bidirectionally to the rest of the level nodes. Its feature embedding is initially computed by averaging all the level node embeddings. It has two purposes: Firstly, it provides some context information to the nodes, as it gathers information from the whole graph. Secondly, it acts as a regularization term for the GAT layer weights, as it is not a real neighbor node. These global nodes are only used during the message passing but discarded once the GNN stage is finished.

For the edge sampling we use a custom approach. We are dealing with unstructured documents with an unknown variability in layouts and we cannot assume any constraint related to the distance between the segments. We define a sampling function that aims at connecting each segment with the rest of the segments that are on the same line or in the adjacent ones: an edge from segment A to

segment B is created if the vertical distance between their centers (C) is less than the height (H) of segment A by a constant (K) (see Equation 2). In our experiments we set this constant to two.

$$edge_{A-B} = |C_A^y - C_B^y| < H_A * K \tag{2}$$

### 3.5 Recurrent layers

The recurrent layers gather the information about the sequence order that the GNN layers are missing and inject it into the node features. More specifically, we use two bidirectional Gated Recurrent Units (GRUs) [15] with 256 hidden features size. We also considered using Long Short Term Memory layers (LSTM) [26], but as reported in the ablation study of the appendix the accuracy obtained is similar while the GRU layers have less parameters and are faster. The contribution of these layers is analyzed in the experimental section.

### 3.6 Classification head

The classification head takes the output features for each node from the recurrent layer and transforms them into the class probabilities. It consists of one linear layer that generates the logits, followed by a Softmax layer that produces the normalized probabilities.

## 4 Experiments

### 4.1 Dataset

We select one well-known public dataset of purchase receipts, CORD [13], that contains annotations for the segment tagging task in order to compare our model with other approaches. In addition, we include a larger and challenging private dataset to better analyze the performance of the model. Due to space limitation, they are further described with examples in Appendix A.2.

### 4.2 Training and Evaluation details

For all the datasets, the model is trained from scratch for 30 epochs using a batch of 4 documents on each iteration. The selected optimizer is AdamW [32] with an initial learning rate of 3e-4 and a reduction factor of 0.1 in epochs 20 and 25. For the loss function, we use Cross Entropy Loss for CORD dataset and Focal Loss for the private in order to deal with the high class imbalance. We also test the impact of pretraining the model on the private dataset before training on CORD. In this case, the models are finetuned for 1000 steps with batch size of 64, an initial learning rate of 1e-4 and a reduction factor 0.1 in step 900. To reduce the overfitting, we use a dropout of 0.1 for the Transformer encoder and before each GAT layer, and a dropout of 0.2 for the GRU layers and before the final linear layer. For both datasets we sort the segments of each document from top to bottom and from left to right in order to have a consistent ordering for the recurrent layers. The maximum character length for the Transformer encoder is 30, longer segments are truncated. Finally, we convert all the characters of the segments into ASCII characters as described in Section 3.3.

### 4.3 Metrics

We select two well-known metrics for evaluating the accuracy of the model:

- F1 score micro: compute the F1 score using all the samples. In this case, all the samples contribute equally to the result, without considering their category.

- F1 score macro: computes the F1 score per class and then averages them to obtain the final score. This is a more robust metric when dealing with unbalanced datasets.

### 4.4 Results

#### 4.4.1 CORD

First, we use the CORD public dataset to compare SeqGraph against other baseline and state-of-the-art models that perform segment tagging. In this case, as the rest of the methods report the results at entity level, we train and test SeqGraph using the annotations at entity level, grouping together

**Table 1:** Comparison with different state-of-the-art models on the tagging task of the CORD dataset at entity level. We also include the number of parameters, if the model needs pretraining or not, and the modality of the input data ("T/L/I" denotes "text/layout/image").

| Model | Parameters | Pretrained | Modality | CORD F1 micro |
|---|---|---|---|---|
| BERT$_{BASE}$[24] | 110M | yes | T | 89.68 |
| RoBERTa$_{BASE}$[25] | 125M | yes | T | 93.54 |
| BROS$_{BASE}$[1] | 110M | yes | T+L | 95.73 |
| LiLT$_{BASE}$[33] | - | yes | T+L | 96.07 |
| TILT$_{BASE}$[22] | 230M | yes | T+L+I | 95.11 |
| LayoutLMv2$_{BASE}$[18] | 200M | yes | T+L+I | 94.95 |
| DocFormer$_{BASE}$[4] | 183M | yes | T+L+I | 96.33 |
| LayoutLMv3$_{BASE}$[3] | 133M | yes | T+L+I | 96.56 |
| BERT$_{LARGE}$[24] | 340M | yes | T | 90.25 |
| RoBERTa$_{LARGE}$[25] | 355M | yes | T | 93.8 |
| BROS$_{LARGE}$[1] | 340M | yes | T+L | 97.4 |
| FormNet[34] | 345M | yes | T+L | 97.28 |
| TILT$_{LARGE}$[22] | 780M | yes | T+L+I | 96.33 |
| LayoutLMv2$_{LARGE}$[18] | 426M | yes | T+L+I | 96.01 |
| DocFormer$_{LARGE}$[4] | 536M | yes | T+L+I | 96.99 |
| GraphDoc[9] | 265M | yes | T+L+I | 96.93 |
| LayoutLMv3$_{LARGE}$[3] | 368M | yes | T+L+I | **97.46** |
| PICK[2] | 68M | no | T+L+I | 95.81 |
| SeqGraph (ours) | **4M** | no | T+L | 96.36 |
| SeqGraph pret (ours) | **4M** | yes | T+L | 97.23 |

the segments that belong to the same entity and using the minimum rotated rectangle as the entity bounding box. The results are reported in Table 1. For all the rest of the methods (except PICK), even their base versions have more than 100 million parameters, while the proposed method hardly reaches the 4 million. Despite this huge difference, SeqGraph outperforms almost all the base versions and most of the large versions of the other state-of-the-art methods with 96.36%. It stays just 1% below the best result achieved by LayoutLMv3 [3] while having almost 100 times less parameters.

Note that almost all the rest of the models are pretrained in other huge datasets before being finetuned in the CORD dataset while SeqGraph and PICK are trained from scratch. Although they are trained in an unsupervised way and in other tasks, this pretraining impacts a lot in the text feature extraction, especially for datasets like CORD with limited training information, where many words that appear in the test set might not appear in the training set but could have been learnt during the pretraining. In order to test this impact in our model, we try pretraining it on the private dataset, even though it has less than 10 thousand single page documents while usually the models are pretrained using millions (for instance LayoutLMv3 uses 50 million pages). As it can be seen in Table 2, this pretraining improves the results of the model, reaching 97.23% and reducing the gap between SeqGraph and LayoutLMv3 to 0.23%.

Another important point that should be taken into account is that the rest of the presented methods are purely or mostly based on Transformer architectures operating at segment level, so they need to define a sequence limit and, therefore, they suffer from the sequence truncation problem. In addition, this sequence limit must be chosen carefully, as the computational complexity increases quadratically with it [7]. However, SeqGraph and PICK do not suffer from this issue, so they do not have a sequence limit. In the CORD dataset this is not a problem, as the number of segments per document is low on average, but for other datasets such as the private receipts dataset, where the receipts may contain hundreds of segments this limitation would impact in the accuracy and could cause the loss of relevant information.

We also extract the results at segment level for SeqGraph (from scratch and pretrained versions) and PICK (see Table 2). Again, SeqGraph outperforms PICK, with a higher difference in this case (almost 2%), and with the pretraining our model improves by 0.47%.

**Table 2:** Results on the tagging of the CORD dataset at word level.

| Model | Parameters | Modality | CORD F1 micro |
|---|---|---|---|
| PICK[2] | 68M | T+L+I | 92.87 |
| SeqGraph (ours) | 4M | T+L | 94.61 |
| SeqGraph pret (ours) | 4M | T+L | 95.08 |

### 4.4.2 Private dataset

For the next experiment, we train and evaluate the proposed model on the segment tagging task of the private dataset. The model is trained following the procedure specified in Section 4.2. We compare our model against PICK [2], as it also performs exclusively the tagging task, it operates at segment level, and it has several similarities with SeqGraph, such as the character encoder for the text features or the combination of GNN and recurrent layers. The PICK model is trained and evaluated using the official repository and the default configuration. Both models were trained on a machine with one NVIDIA Tesla V100 GPU, 64 GB of RAM, and 1 Intel(R) Xeon(R) Gold 6142 CPU.

The micro and macro F1 score for both methods are presented in Table 3, along with the number of parameters (in millions), the modality of the input data ("T/L/I" denotes "text/layout/image"), and the time taken for the whole training process. Note that the proposed method has 17 times less parameters than PICK and that, unlike PICK, it does not use the image as an input source. Despite this, it can be observed that SeqGraph outperforms PICK in both metrics and specially in the F1 macro, where it improves more than a 1%. These results demonstrate that the image does not provide additional relevant information to the one extracted from the text, the layout, and the segment order.

**Table 3:** Results on the segment tagging task of the private receipts dataset. We also include the number of parameters, the modality of the input data ("T/L/I" denotes "text/layout/image"), and the total training time.

| Model | Parameters | Modality | F1 micro | F1 macro | Training time |
|---|---|---|---|---|---|
| PICK[2] | 68M | T+L+I | 96.99 | 93.27 | 33h |
| SeqGraph (ours) | 4M | T+L | 97.47 | 94.51 | 1h20m |

Regarding the training time, for the same number of epochs, PICK was trained in 33 hours (1 hour per epoch) while SeqGraph was trained in 1 hour and 20 minutes (less than 3 minutes per epoch). Some of the causes of this overwhelming difference are the heavy image feature extraction done by PICK or the fact that the recurrent layers of PICK process the sequences at character level. An ablation study provides further discussion in Appendix A.3

## 5 Conclusions and Future Work

In this work we have addressed the problem of text segment tagging on unstructured documents. We believe that the existing state-of-the-art models are unnecessarily huge, with an overwhelming number of parameters. Furthermore, most of them are based on Transformer architectures, suffering from the sequence truncation problem and not taking advantage of the sparse nature of the use case. To overcome these limitations, we have proposed SeqGraph, a new model which optimizes the feature extraction stage and mixes GNNs and RNNs to efficiently and effectively solve the segment tagging problem. We have demonstrated its capabilities by testing it on the CORD dataset, where it achieves state-of-the-art results while reducing the number of parameters between 100- and 200-times compared to its competitors. In the benchmark against PICK [2], we have also demonstrated that the image features are not essential for this task and that they do not provide additional relevant information that can be added to the one extracted from the OCR text segments.

Future work will focus on improving the performance of the model trying to mitigate the bottlenecks. For instance, injecting positional embeddings into the node features, to see if the sequence information can be extracted by the GAT layers and thus removing the recurrent layers, which are computationally heavy. Another research line is extending the capabilities of the model to cover also segment grouping and entity linking tasks, evolving into an end-to-end information extraction model.

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

# A   Appendix

## A.1   Analysis of the text feature extraction methods

As it is described in Section 3.3, we can group the text feature extraction methods in two main categories: the ones that extract the features attending to its semantic meaning and the ones that extract the features attending to its composition.

Feature extraction based on the semantic meaning of the features has some important drawbacks listed below:

- The words that are not in the dictionary will get assigned a useless embedding, so the model will not perform well on unseen data or on noisy data (for instance due to parsing errors in the OCR module).

- The size of the dictionary must be huge to include as many words as possible, and so does the size of the dataset for generating it. This size problem worsens when working with multilingual data.

- It is very prone to overfitting, especially for the words that are less frequent.

- Problems when dealing with wide ranges of numbers. For instance, when working with prices, the model cannot extract general rules for them and needs to treat each number as an independent word, while it is not feasible to include all the possible numbers in the dictionary.

Some of these weak points can be partially mitigated by decomposing the text in known character grams and extracting the features from them [1, 30]. This variant can improve the performance over noisy and unseen data and reduce the overfitting. Nevertheless, this strategy can increase even more the size of the dictionary and the models are still very sensitive to the noise and the unseen data.

On the other hand, extracting the features attending to the text composition has the following advantages:

- The size of the dictionary is drastically reduced. For instance, if considering only the ASCII characters the length would be only 128.

- The models are more robust to unseen or noisy data, as even if some characters are missing or they are different, it can find relationships between the rest of them.

- The models tend to analyze the words at a lower level, without attending too much to the semantic meaning and finding more general rules, which reduces overfitting.

- The numbers range problem is eliminated, as it can find general rules to group all the segments of the same type under the same meaning. For instance, the model could learn that when a digit is followed by a dot and then by other digits, the segment is a price without having to analyze all the possible numbers.

- These previous advantages also impact on reducing the amount of data required for pretraining the embedding layer.

## A.2  Datasets

### A.2.1  CORD

Consolidated Receipt Dataset [13] is composed of 1000 Indonesian purchase receipts which contain images and box/text annotations for OCR, and multi-level semantic labels for semantic parsing and relation extraction tasks. In the ground truth, each segment is associated with the $category$ field and the $group\_id$ field for joining the segments at entity level. It contains 30 different categories. The samples are split into 800 for train, 100 for dev (validation), and 100 for test.

### A.2.2  Private dataset

For effectively evaluating the capabilities of the model, we propose an internal challenging dataset composed of 8814 purchase receipt images from 5 countries: Germany, Italy, France, Mexico, and Brazil. Receipts vary widely in height, density, and image quality. They may contain perspective artifacts, 3D rotations and all kinds of wrinkles. Each receipt has all its text segments annotated. The available annotated information for each text segment is the rotated bounding box, the text, the entity category, and the product ID (in case the segment belongs to a product cluster). About the entity categories, there are 21 types of different entities, some of them at receipt level ($purchase\_date$, $purchase\_time$, $total\_value$,...) and others at purchase item level ($item\_description$, $item\_code$, $item\_value$, ...).

The dataset also contains the receipt region annotation. We have cropped the images, filtering the segments that are outside the receipt, and shifting the coordinates of the remaining segments to the cropped pixel space. Finally, we split the dataset in training, validation and test sets using a ratio of 70/10/20. In Figure 5 we present some examples. We also overlay the ground-truth labels, where the boxes with the same color belong to the same category. Note that this dataset is more challenging

than CORD in the number of samples, languages, high imbalance in the classes (especially for 'other', i.e. text not belonging to targeted classes) and that the number of segments can vary from several to hundreds from one receipt to another. Furthermore, the layouts may vary highly intra- and inter-retailers and there are a large number of them (hundreds per country). Finally, note also that the quality of the receipts related to paper and printing defects and image capture is worse than in CORD, which means injecting more noise and variability into the input data.

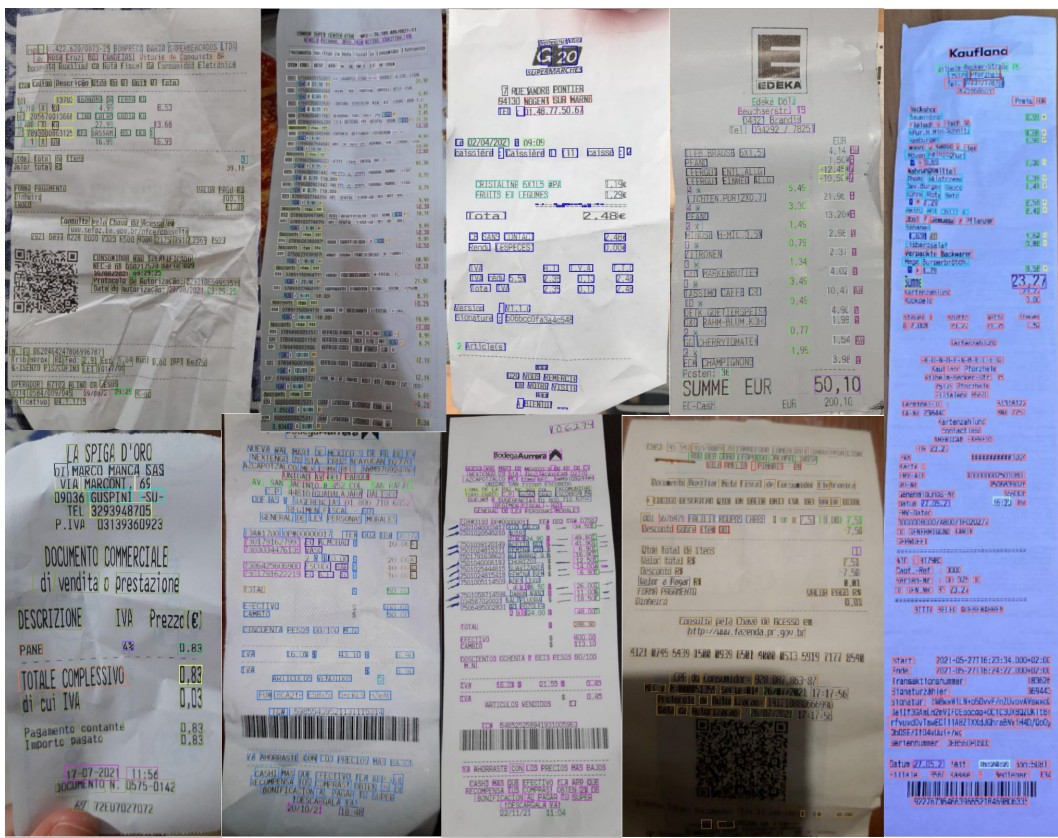

**Figure 5:** Examples from the private dataset of receipts including ground truth labels. The color of the bounding box denotes its category. On each image, multiple text segments can belong to same category.

### A.3 Ablation study

We analyze some design choices and their impact on the model accuracy and on the number of parameters. All these experiments are performed using the private dataset and the results are gathered in Table 4.

First, we study the differences between using GRU [15] and LSTM [26] as the recurrent layers. The experiment shows that the results slightly improve when using GRU layers, while reducing the number of parameters of the model by 0.6 million.

Next, we want to analyze the contribution of each source of input information. We start with the sequential information, which is gathered by the recurrent layers. Thus, we remove the recurrent layers, connecting the output of the GNN directly to the final linear layer (SeqGraph w/o RNN in Table 4). As can be observed, there is a drop of 1% in F1 score micro and almost 3% in F1 score macro. However, note that there is also an important drop in the number of parameters, almost 50%. In order to compensate for this drop, the number of GAT layers is increased from 3 to 5, and the number of heads of each layer from 4 to 8. With this variant (SeqGraph w/o RNN extended) the drop is halved, but it is still important for the F1 score macro. Therefore, we can conclude that the sequential information is relevant for this task and that it cannot be fully replaced by the layout features.

**Table 4:** Ablation study on the proposed method. SeqGraph is the baseline model; SeqGraph LSTM replaces the GRU layers by LSTM ones; SeqGraph w/o RNN removes the recurrent layers; SeqGraph w/o RNN extended removes the recurrent layers and add more GAT layers to compensate the drop of parameters; SeqGraph w/o RNN & layout removes the recurrent layers and the layout features; and SeqGraph w/o text removes the text feature extraction module.

| Model | Parameters | Modality | F1 micro | F1 macro |
|---|---|---|---|---|
| SeqGraph | 4M | T+L+S | 97.47 | 94.51 |
| SeqGraph LSTM | 4.6M | T+L+S | 97.40 | 94.39 |
| SeqGraph w/o RNN | 2.3M | T+L | 96.33 | 91.56 |
| SeqGraph w/o RNN extended | 6.5M | T+L | 96.84 | 92.81 |
| SeqGraph w/o layout | 4M | T+S | 97.22 | 94.18 |
| SeqGraph w/o RNN & layout | 2.3M | T | 93.98 | 87.99 |
| SeqGraph w/o text | 2.4M | L+S | 95.13 | 88.22 |

The next source of information considered is the layout, embedded in the coordinates of the segment bounding boxes. We remove this information from the node features, maintaining only the text ones (SeqGraph w/o layout). Surprisingly, the drop in performance is almost null, 0.25 in micro and 0.33 in macro metrics. Nevertheless, note that the layout features are also employed during the edge sampling step for finding the neighbors of each node, so we believe that this information embedded in the graph structure, together with the sequence information, is mitigating the suppression of the features from the nodes.

We also try removing both the sequential and the layout information (SeqGraph w/o RNN & layout). In this case, the drop in performance is huge, 6.6% in the macro metric. This demonstrates that, although each of these sources contains some exclusive relevant information, most of it is shared by both.

Finally, we test a version of the model where we remove the text information from the node features (SeqGraph w/o text). As expected, the F1 score macro highly decreases (more than 6%), demonstrating that the text features are the most important ones, but that they need to be complemented with other sequential and/or layout features.

