# OpenReview forum: "Combining Graph and Recurrent Networks for Efficient and Effective Segment Tagging"
_logconference.io/LOG/2022/Conference — LoG 2022 Poster_

### Official Review · Reviewer_mbZx · 2022-10-18

**Overall Score:** 8
**Confidence:** 4

**Review:**

##########################################################################

Summary: This paper proposes a Transformer-GNN-RNN hybrid architecture for the semantic tagging of documents. More specifically, the method utilizes a Transformer for text feature extraction, a GAT for locally constrained segment interaction, and a GRU for an ordered segment interaction across a document. Through a careful selection of the design choices, the proposed approach achieves state-of-the-art results on the CORD dataset with ≤1% of the parameters to previous approaches.

##########################################################################

Reasons for score:

Overall, I vote for accepting. The paper tackles a practical problem with unique challenges in an effective manner, and comprehensively justifies most of the design choices through an extensive literature survey, discussion, and ablation experiments. My major concern is not evaluating the model in three out of four widely used benchmarks (see Weaknesses). I hope that the authors address the concern in the rebuttal period.

##########################################################################

Strengths

- The paper tackles a practical problem of semantic tagging of documents that comes with unique challenges.
- The paper provides a comprehensive review of the related work in Section 2, which helped me to understand the relevant approaches and motivation, and validity of the proposed method.
- Almost every design choice, e.g., GNN-RNN hybrid approach, not using sequence positional encodings, SiLU activation, global document node, and GRU layers, are concretely justified in Section 3 with literature survey, discussions, and experiments.
- In Section 4, the method empirically shows an impressive performance compared to the state-of-the-art approaches with only a tiny fraction (≤1%) of parameters.
- The authors provide a comprehensive ablation study on the design choices.

##########################################################################

Weaknesses

- Prior work including LayoutLMv3 [4] also performs the evaluation in FUNSD, RVL-CDIP, and DocVQA, on top of CORD. Is there any reason that the proposed method is only evaluated in CORD? Since the model architecture is extensively engineered on the CORD dataset, it is easy to overfit the design choices specific to the dataset. If one or two datasets could be added, that would be great to demonstrate the generalizability of the architecture.

Questions and comments

- Line 35-37 and Line 274-275: As far as I know, the number of parameters within a transformer model is not dependent on the input sequence lengths. Perhaps the authors are referring to the quadratic asymptotic computational complexity of self-attention to the input length [1]?
- Line 123: The sentence can be a bit misleading since the paper is not the first to propose a GNN solution. Perhaps rewriting it as “… we focus on a GNN-based model due to these reasons” could be more clear.
- Line 128: May I ask for references on global modeling? To my knowledge, GNNs without modifications face challenges in modeling global interactions [2].
- In Section 3.3, is the Transformer for text encoding trained from scratch?
- In Section 3.5, how are the output nodes of GNN ordered into a sequence to be used as an input to RNN? Is it through raster scan ordering?
- Line 187: The word “positional feature” can be confusing since the previous paragraph explains that sequence positional feature is not used. Maybe use “bounding box features” or “bounding box positional features” instead?
- The presentation of Eq. (1) is a bit cluttered and can be improved.
- In the last line of Eq. (1), the attention coefficient \alpha_ij^{(l)} is placed over the sum symbol. Is this a typo?
- In Section 4.1, it would be helpful to understand the experimental setups if the sizes of the datasets (CORD and private) are provided.
- Line 235: The FUNSD dataset is mentioned but not used in the paper.
- Line 252: The name of the model SeqGraph is not introduced in the previous parts of the paper. Perhaps it would be better to introduce it in the Methods section in advance.

##########################################################################

[1] Choromanski et al., Rethinking Attention with Performers (2020)

[2] Wu et al., Representing Long-Range Context for Graph Neural Networks with Global Attention (2022)

[3] Huang et al., LayoutLMv3: Pre-training for Document AI with Unified Text and Image Masking (2022)

---

### Official Review · Reviewer_U123 · 2022-10-22

**Overall Score:** 3
**Confidence:** 4

**Review:**

This problem examines the task of information extraction on documents when dealing with unstructured documents. The paper presents a model SeqGraph, that combines Transformers for text  feature extraction, and Graph Neural Networks and recurrent layers for segments interaction, for segment tagging.

Strong Points:
The paper is overall well-written and well-structured. Further, the motivation provided for using GNNs is sufficiently summarized with limitations of existing works described. The experiments on training time is interesting. In the experiments section, there are numerous baselines provided for the different types of vanilla pretrained language models.

Weak Points:
Introduction and Related works while well organized are missing mention of some important state-of-the-art GNN models. For example, recent models utilize powerful hierarchical attention mechanisms:
[1] Bi-Level Attention Graph Neural Network (ICDM 2021): https://ieeexplore.ieee.org/abstract/document/9679017
[2] Heterogeneous Graph Attention Network (WWW 2019): https://arxiv.org/pdf/1903.07293.pdf

In the methodology, it would be helpful to include the GNN loss function formula details explicitly.

In the experiments/ablation studies section, it would also help to see different combinations of GNN model layers and activation functions e.g., GCN, softmax(), etc., to show a more systematic approach in motivating why the proposed GNN architecture is used (over others).
Further, experiment results performance gain seems to be minimal in comparison to the baselines. Evaluation on more datasets are also needed to validate that there are consistent results.

---

### Official Review · Reviewer_PpDN · 2022-10-22

**Overall Score:** 8
**Confidence:** 4

**Review:**

** Summary**
The authors propose a sequence tagging method using GNNs and RNNs. Existing approaches are primarily based on Transformer based large language models (LLMs), as a result they are huge in terms of number of parameters. Whereas the proposed approach called SeqGraph is 100 times smaller while retaining similar performance.

**Positives**
1. The paper is very well written with multiple figures. Particularly, Figure 1 was really informative to set the task context, it provides much needed context and sets the practical aspects of the problem clear. I appreciate the authors for including this in the paper.

2. I feel SeqGraph has strong technical contribution and will be of interest to both NLP and GNN domains.

3. Strong experiments and results further provides sufficient evidence for contributions.

4. I appreciate the authors for accepting the limitations of their approach upfront, for example "GNNs do not consider the positions of the segments".

**Limitations**
Although I liked the paper very much, I feel a few small incorporations will improve the paper quality.
1. Authors mentions as the motivation for choosing GAT as "as the one that best suits our needs." I would really appreciate if the authors could explain a little why the choice of GAT makes sense. Maybe estimating the edge weights helps ignore the unrelated segments?

2. This is related to previous comment, in the ablation, I would prefer a comparison of other GNN archs such as GCN, GIN, etc. However, I am okay with the current ablation too.

3. The edge sampling process will result in multiple disconnected components, did the authors perform any steps to handle this?

4. I would have loved to see experiments on another public dataset, however I understand the limitations regarding it.

---

### Official Review · Reviewer_vQaa · 2022-10-27

**Overall Score:** 6
**Confidence:** 3

**Review:**

1. Contribution Summary

This paper proposes to combine Transformer neural network, Graph Neural Network and Recurrent Neural Network to do the OCR-based Segment Tagging task. The three technique components are popular and powerful individually, and they have been successfully used in different areas separately. It is expected combining them together will give better performance.

2. **Strong** and **Weak** points.

**Strong**, the paper has good writing and is well-presented. Delivering each technique component clearly and coherently. The experiment is thoroughly presented.

**Weak** There are several weak points that I heavily concern.

* **Efficient and Effective Segment Tagging claim**,  the authors claimed their model is efficient and only has 4M param size. To me, however, such claim might be over-claimed. In the paper, the author compared their method with SeqGraph, BERT, RoBERTa and other methods, and show their method has minimal parameter size. The other methods may not directly focus on the same task, so directly comparing with them in terms of parameter size, regardless of task, is relatively unfair. Moreover, if the authors want to sell the "efficient and effective point", they also need to some studying on their work's inference time test and report it. I personally think adopting two bidirectional Gated Recurrent Units will not produce test efficiency.

* **Imbalanced Technique Representation**, the author invested large area and discussion on introducing their overall general idea. When it comes to the core technical part, however, the corresponding discussion is minimized. For example, Transformer is an important part, but I can just see brief introduction in line 169-170. Each math number meaning in Eq. (1) is missing as well. Such imbalanced writing makes reviewers hard to follow their core contribution.

* **Transformer+GAT+GRU**, again, I personally don't think bi-GRU is that efficient, did the authors to replace the bi-GRU with Transformer architecture so that the attention can be computed in parallel?

3. **Questions to Authors**

See the **Weak Points**.

---

### Meta-Review · Area_Chair_fq6o · 2022-11-20

**Confidence:** 5
**Recommendation:** Accept

**Meta Review:**

The paper describes a GNN-based approach for segment tagging. This is primarily an application paper that develops a purpose-built approach for the task by combining graph and sequence models. The paper is well-written and the approach/choices are well-motivated. Experiments show that this approach performs well, outperforming all but the current SOTA approached based on layout-aware LMs. The weakness of the approach is that it has been only evaluated on a single dataset. As such, it is not clear how its heuristics around node/edge construction will generalize to other datasets/domains. Overall it is a solid application paper and the majority of the reviewers support its acceptance.

---

### Decision · Program_Chairs · 2022-11-23

Accept (Poster)